# MuSK Myasthenia Gravis—Potential Pathomechanisms and Treatment Directed against Specific Targets

**DOI:** 10.3390/cells13060556

**Published:** 2024-03-21

**Authors:** Edyta Dziadkowiak, Dagmara Baczyńska, Marta Waliszewska-Prosół

**Affiliations:** 1Department of Neurology, Wroclaw Medical University, Borowska 213, 50-556 Wroclaw, Poland; edyta.dziadkowiak@umw.edu.pl; 2Department of Molecular and Cellular Biology, Wroclaw Medical University, Borowska 211A, 50-556 Wroclaw, Poland; dagmara.baczynska@umw.edu.pl

**Keywords:** myasthenia gravis, MuSK, neuromuscular disease, autoimmune

## Abstract

Myasthenia gravis (MG) is an autoimmune disease in which autoantibodies target structures within the neuromuscular junction, affecting neuromuscular transmission. Muscle-specific tyrosine kinase receptor-associated MG (MuSK-MG) is a rare, often more severe, subtype of the disease with different pathogenesis and specific clinical features. It is characterized by a more severe clinical course, more frequent complications, and often inadequate response to treatment. Here, we review the current state of knowledge about potential pathomechanisms of the MuSK-MG and their therapeutic implications as well as ongoing research in this field, with reference to key points of immune-mediated processes involved in the background of myasthenia gravis.

## 1. Introduction

Myasthenia gravis (MG) is an autoimmune disease of the postsynaptic part of the neuromuscular junction (NMJ). Immunologically, MG is a heterogeneous group caused by different, pathogenic antibodies against important synapse proteins. These antibodies include Ig1 or Ig3 class antibodies against acetylcholine receptor (AChR), Ig4 class antibodies against muscle-specific kinase receptor (MuSK), and antibodies against lipoprotein receptor-related protein 4 (LRP4). Patients with MG have a similar clinical presentation, but the immunopathology is unusually heterogeneous [1,2,3,4].

Approximately 5–8% of myasthenia gravis patients are positive for antibodies against muscle-specific tyrosine kinase receptors [5,6]. Its prevalence varies between countries and ethnic groups. Higher rates of MuSK-MG patients are observed in southern Europe, with a pronounced prevalence in females, who account for more than 70% of patients. The disease has an earlier age of onset, with a peak incidence in the latter part of the third decade of life, and rarely occurs after the age of 70 [5]. In contrast to AChR-MG, no significant thymus alterations, such as thymic hyperplasia, have been reported in MuSK-MG patients [7]. Furthermore, it is postulated that thymectomy does not improve clinical outcomes in these patients [8].

MuSK-MG is a rare, often more severe subtype of the disease with different pathogenesis and specific clinical features. MuSK-MG usually has an acute onset, involving predominantly facial and bulbar muscles. Symptoms usually develop progressively, over the course of several weeks. Initial respiratory crises are common. The disease can lead to generalized muscle weakness to the stage of muscle atrophy. The muscle groups mainly involved are facial muscles and the tongue. Severe skeletal muscle involvement can also be confirmed [4,5,6,7]. The atypical onset of the disease, such as ocular involvement, lack of variable symptoms, failure of acetylcholinesterase inhibitors, and negative electrophysiological studies, impede the diagnosis of MuSK-MG [5,9,10].

Here, we highlight the immunological mechanisms of the MuSK subtype of MG, the commonality with other autoimmune nervous system disorders, and note the most recent approaches to treatment.

## 2. Methods

The authors conducted a literature search focused on the topic of the pathomechanisms and treatment of MuSK myasthenia gravis. The key search terms applied in PubMed via MEDLINE were “myasthenia gravis” or “MG” or “MuSK MG” and “pathomechanisms” and “immunology” and “treatment” (Figure 1). The online search covered the publication period from database inception, i.e., 2010, until 31 December 2023. Reviews and research studies, classified according to their relevance, were initially included, with the subsequent exclusion of conference abstracts and papers written in languages other than English. In addition, reference lists from the eligible publications were searched for their relevance to the topic.

## 3. MUSK: From Gene to Functions

Signal transmission is involved between the motoneuron and the muscle fiber muscle, a specialized structure called the neuromuscular synapse or neuromuscular junction. The motoneuron, together with the muscle fiber (or a group of fibers of the same type) innervated by it, forms a motor unit.

In an NMJ, one can distinguish three essential elements:The presynaptic part, including the motoneuron endings;The synaptic gap into which synaptic vesicles are secreted from the motoneuron axon and from which the neurotransmitter, acetylcholine, is released;The postsynaptic region situated on the sarcolemma, which contains acetylcholine receptors. The binding of acetylcholine by these receptors initiates a cascade of events leading to muscle contraction [11,12].

Formation of NMJs involves a complex signaling process, both spatially and temporally, between motoneurons and muscle myotubes, the end result of which is the clustering of acetylcholine receptors (AChRs) on the postsynaptic side of the junction and a differentiated nerve terminal on the presynaptic side. The key proteins in NMJ formation include a neuronally derived heparan-sulfate proteoglycan, agrin, and three muscle proteins: downstream of kinase-7 (Dok7), low-density lipoprotein receptor-related protein-4 (LRP4), and rapsyn [13,14]. LRP4 serves as a cis-acting (in muscle) transmembrane ligand for MuSK; agrin acts as an allosteric regulator of LRP4’s interaction with MuSK; Dok7 functions as a cytoplasmic activator of MuSK, whereas rapsyn binds directly to AChR to facilitate its clustering [15,16]. Muscle-specific kinase was identified as a postsynaptic integral membrane protein playing a crucial role in the development of the neuromuscular junction synapse (Figure 2). The absence of NMJs is lethal. The inability to form or maintain normal NMJs results in neuromuscular transmission pathologies such as myasthenia gravis and congenital myasthenic syndromes (CMS).

MuSK was described for the first time as a novel Trk-related receptor tyrosine kinase (RTK) enriched in the electric organ of *Torpedo californica*, a species of electric ray in the family *Torpedinidae* [15,16,17,18]. Human gene coding MuSK is located on chromosome 9q31.3 and consists of 11 constitutive and five alternative exons. Six transcript variants of MuSK have been identified due to alternative splicing [19]. Initially, MuSK expression was considered tissue-specific and limited to skeletal muscle cells. However, more detailed investigations have shown the highest level of its transcripts in the small intestine and similar to skeletal muscle expression in the testis, bladder, and lung. The expression in brain tissue is extremely low. However, it is detectable in some brain regions, especially the epithalamus. The detection of MuSK transcripts in mouse and human vascular leptomeningeal cells (VLMCs) additionally indicated alternative functions of MuSK in both neuronal and non-neuronal cells [20]. Valenzuela et al. noticed the high transcript expression of two main MuSK isoforms during the early embryonic development of rat myotome, which then persisted in time of skeletal muscle formation. However, its mRNA has dramatically decreased after birth [21]. Despite many similarities between rodent and human MUSK genes, there are also differences. Nasrin et al. detected three alternative splicing isoforms of MuSK transcripts in human skeletal muscle.

Furthermore, they have observed unique for the human gene, alternative exon 10 defective skipping. The exclusion of this exon from mRNA is more frequent in undifferentiated and poorly differentiated human myoblast and myogenic cells than in skeletal muscle [19]. Interestingly, two MuSK isoforms (one identical to the skeletal muscle variant) are expressed in the brain. It was shown that hippocampal MuSK isoforms play a crucial role in cholinergic response and help memory formation [20]. These data obtained on animal models confirm observations of MG patients with memory deficits [22]. Human MuSK is a transmembrane glycoprotein of type I and consists of several extracellular domains, a single transmembrane helix, and a cytoplasmic tail with a tyrosine kinase domain (TKD, Figure 2). The N-terminal fragment of MuSK includes a signal peptide, followed by three immunoglobulin-like domains (Ig), and a frizzle-like cysteine-rich domain (Fz-CRD). The first two Ig are crucial for lipoprotein receptor-related protein 4 (LRP4) interactions as well as homodimerization. The central point of LPR4— the binding site on Ig1—is determined by Ile96, and its mutation decreases the interaction of MuSK with LRP4 [20]. These interactions can be enhanced by agrin. Thus, agrin plays a role as an allosteric and paracrine regulator, and its presence is not necessary for MuSK activation [23]. Furthermore, LRP4 binding to MuSK impacts the hydrophobic surface situated opposite to Ile96 and aids direct interactions between Leu83 and Met48, which leads to Ig1- dimerization and autophosphorylation reactions [24]. The role of Fz-CRD is believed to be essential for MuSK activation by Wnt signaling proteins in the lack of agrin [18].

The transmembrane domain is linked with TKD by a cytoplasmic juxtamembrane segment. The autophosphorylation of Tyr553 within this segment creates the docking site for the cytoplasmic protein Dok7. Similar to LRP4, Dok7 binding increases the strength of MuSK dimerization, which is crucial for further activation of its kinase domain [25,26]. For full activation, two (Tyr754, Tyr 755) of three (Tyr, 750, Tyr754, Tyr 755) tyrosines within the activation loop of TKD need to be phosphorylated in the established order (Figure 3). In contrast, the lack of phosphate groups on these tyrosines autoinhibits the activity of MuSK [27]. Simultaneously with autophosphorylation of the activation loop, Dok7 is phosphorylated at Tyr369 and Tyr406, which permits the binding of the adapter molecule Crk and NMJ formation [28].

### MuSK: From Gene to Disease

Currently, MuSK-MG is only diagnosed by detecting the autoantibodies against MuSK in patients’ serum or plasma. AChR-MG, as well as LPR4-MG, correlates with increased levels of IgG1 and IgG3 autoantibodies [29]. In contrast, antibodies against MuSK belong mainly to the subclass IgG4. These IgG4s can block direct interaction between MuSK and complex collagen Q-AChR and inhibit agrin-induced phosphorylation of MuSK, leading to attenuation of AChR clustering [30,31]. IgG4 antibodies do not affect MuSK phosphorylation in cases of a lack of agrin stimuli. The same final effect of clustering of AChR inhibition can be achieved by the less common IgG1-3 anti-MuSK antibodies. In contrast, IgG1-3 can act independently on agrin stimuli and leads to increased MuSK phosphorylation. Thus, both inhibition and overphosphorylation of MuSK can attenuate AChR clustering and lead to disorders in NMJs [31,32]. The pathogenicity and acute course of the disease result directly from the quantitative composition of individual IgG fractions, as well as the reduction of their galactosylation levels [33,34].

It has been shown that HLA class II DR14, DR16, and DQ5 alleles are dominant in MuSK-MG and can be predisposed to the production of autoimmunogenic IgG4 antibodies. On the other hand, HLA-DR13 can be beneficial and protects against pathological IgG4 [5,6,35,36,37].

The role of cellular immune response in MuSK-MG is still unknown. The contribution of Th1 and Th17 cells in this disease subclass has been postulated. The unclear mechanism of the regulation of the activity of T cell subsets causes additional constraints in the development of effective therapies [38]. Myasthenia gravis is associated with different circulating miRNA profiles [39]. However, their role, especially in the pathogenesis and development of MuSK-MG, is poorly studied. Elevated miRNAs in the serum of patients with MuSK-MG include miR-151a-3p, let-7a-5p, let-7f-5p, and miR-423-5p. Furthermore, downregulation of miR-210-3p and miR-324-3p has been found in the plasma of MuSK-MG relative to healthy controls. Despite altered microRNA expression in MuSK-MG patients’ PBMC, its targets and the signaling pathways that may play key roles in the development of the disease remain unclear. For comparison, the state of knowledge about AChR-MG is much more advanced, and increased expression of miR150-5p, as well as miR21-5p, is associated with the differentiation and cellular response of T cells. Moreover, miR30e-5p has been found to be a promising predictive biomarker for the disease. It is noteworthy, however, that studies remain very limited and require proper grouping of patients. Circulating miRNAs are not used in routine clinical practice for the diagnosis of MG [40,41,42].

Although MuSK-MG is classified as an autoimmunological disease, some mutations of the MuSK gene may lead to similar symptoms. According to the National Center for Biotechnology Information data (NCBI, www.ncbi.nlm.nih.gov), 52,922 different mutant variants of the *MUSK* gene were detected. The mutations that abolish the protein’s activity and functions are usually lethal. Crucial are the amino acids responsible for interactions with LRP4 (Ile96, Leu83, and Met48), autophosphorylation of the tyrosine kinase domain (Tyr 553), and Dok-7 interaction (Val790, Met605, Ala727). Other amino acids can result in diminished expression of MuSK (c220insC). Similarly, dysfunctional mutations in all NMJ proteins can lead to congenital myasthenic syndrome [43,44,45].

## 4. Specificity of Neurophysiological Diagnostic Tests

Extended neurophysiological assessment in patients with myasthenia gravis includes repetitive nerve stimulation (RNS), quantitative EMG (QEMG), single fiber electromyography (SFEMG), and electromyography (EMG) with nerve conduction study.

The SFEMG test was developed in the 1970s by Ekstedt and Stålberg [46,47]. Initiating research into quantifying muscle fatigue, Ekstedt and Stålberg developed a multi-electrode electrode for recording action potentials from single muscle fibers (SFAPs), which was inspired by the electrode used by Buchthal et al. [48]. The criterion for SFAPs was a rapidly increasing positive–negative peak of constant shape with successive discharges. Differences in the timing of SFAPs resulted in a ‘jitter phenomenon’, which was attributed to differences in the time at which muscle action potentials are initiated at the motor endplate. This multi-electrode electrode was also used in the analysis of the propagation velocity of individual muscle fibers. SFEMG allows quantitative measurement of, among other things, the variability of neuromuscular transmission (jitter) during successive discharges of an individual muscle fiber action potential. The test is performed by activating the muscle with a voluntary contraction, recording potentials from the muscles of the face (orbicularis oculi, frontalis) or upper extremity (extensor digitorum brevis). The average jitter is automatically calculated from 20 pairs of potentials recorded from several electrode positions. An increase in jitter with any intermittent impulse blocking indicates a significant disturbance in neuromuscular transmission. Measurement of jitter has proven to be the most sensitive method in detecting this type of pathology. This method is highly sensitive but not completely specific for myasthenia gravis. Abnormal SFEMG findings, i.e., increased jitter and blocking, also occur in Lambert–Eaton myasthenic syndrome, as well as in some other neuromuscular diseases such as amyotrophic lateral sclerosis, other neurogenic lesions, and some myopathies (including progressive external ophthalmoplegia, muscular dystrophies, and myositis) [47,48,49]. In myopathy, reinnervation, fiber splitting, and denervation in the late stage of fibrosis may be responsible for the increase in fiber density (FD) parameters. However, jitter is increased in most patients with mitochondrial diseases that primarily affect the extraocular muscles. In progressive extraocular ophthalmoplegia (PEO), as in MG, abnormal jitter can be obtained, making it impossible to diagnose these disorders with SFEMG alone [49,50,51,52,53,54,55].

The results of RNS testing in myasthenia gravis patients with anti-MuSK antibodies are similar to those from SFEMG testing, showing a higher rate of positive results (sensitivity 86%) for facial muscle testing compared to MG cases with AchR antibodies (sensitivity 82%). This reflects the greater propensity for facial muscle involvement in these MuSK antibody-positive cases and highlights the importance of including facial muscles in RNS protocols when assessing these patients [56]. However, Padua et al. conducted a study in a group of patients with seronegative myasthenia gravis (SNMG) which distinguished patients with (USK(+)) and patients without (MUSK(−)) anti-MuSK antibodies. The authors revealed that the RNS test was abnormal in significantly more MUSK(−) than MUSK(+) patients (*p* < 0.00001), while MuSK- patients had a more severe neurophysiological pattern with SFEMG [57].

Comparisons between RNS and jitter analysis, between MuSK-MG and AchR-MG patients, have shown that RNS is less sensitive (52%) in MG patients with antibodies to muscle-specific kinase compared with MG patients with antibodies to the acetylcholine receptor (93%) (*p* < 0.01) [58,59]. Nemoto et al. found positive jitter in 93% of patients with AchR antibodies, but only in 50% of patients with MuSK antibodies, and the range of jitter was greater in AchR-antibody patients versus AchR-negative patients (MCD: 76 μs in patients with AchR antibodies, 36 μs in patients with MuSK antibodies) [60]. In contrast, Nikolic et al. found no significant difference in pathological jitter detection between the two subtypes of MG patients (90% in patients with MuSK antibodies vs. 93% in patients with AchR antibodies, *p* > 0.05) [61]. However, the extent of jitter may be partly due to the severity of the dysfunction in different muscles. Kuwabara et al. found abnormal jitter in the extensor digitorum communis (EDC) muscle in only one of three MuSK-positive patients, but all three had increased jitter in the frontalis muscle [62]. In contrast, all AchR-positive patients (*n* = 11) showed equally abnormal jitter in both muscles. Similar results were reported in a different study by Farrugia et al., where the greater of patients with MuSK antibodies (*n* = 13) had normal jitter in the EDC notwithstanding abnormal jitter in the orbicularis oculi muscle [63]. Since patients with MuSK antibodies are thought to have predominant muscle weakness in the bulbar, facial, and neck compared to patients with MG with AchR antibodies, SFEMG should be undertaken in the most apparent muscles when MG with MuSK antibodies is suspected in order to increase sensitivity [58]. In contrast, a similar degree of SFEMG abnormalities was present in proximal muscles between MuSK(+) and AchR(+) patients [64].

Overall, the literature also contains studies that show proximal myopathy is overrepresented in MuSK(+) patients compared to AchR(+) patients. Both MuSK(+) and MuSK(−) patients, in contrast, have mild myopathy with frequent mitochondrial abnormalities [63,64].

## 5. Non-Neurological Manifestations of MuSK-MG

Myasthenia gravis also leads to reduced psychological and social well-being. The literature presents papers on quality of life (QoL) in the MuSK-MG patient population compared to AchR MG patients [65,66,67,68]. To assess health-related QoL, the SF-36 questionnaire and scales are most commonly used: the Hamilton scale for depression and anxiety, the Multidimensional Perceived Social Support Scale, and the Illness Acceptance Scale. In the study by Stankovic et al. [65], QoL scores in the physical domain were indistinguishable in MuSK-MG and AchR-MG patients, while the mental domain and total SF-36 scores were even better in MuSK groups. Social support was better in the MuSK group. The SF-36 total score correlated with anxiety (rho = 0.49, *p* < 0.01), depression (rho = 0.54, *p* < 0.01), and MSPSS (rho = −0.35, *p* < 0.05), and depression was an independent predictor of worse QoL. The authors conclude that, in addition to therapy for weakness, psychiatric treatment and various forms of psychosocial conditioning should form part of regular treatment protocols in MG [65].

## 6. Molecular Commonalities between MuSK-MG and Other Autoimmune Diseases of the Nervous System

The IgG4 autoantibody subclass is implicated in a broad spectrum of more than 12 multisystem or fibroinflammatory autoimmune diseases, referred to as IgG4-related diseases (IgG4-RD) [69,70]. IgG4 neurological disorders (IgG4-ND) are now developing into an immunopathologically distinct spectrum of diseases, as recently indicated, due to their association with pathogenic IgG4 antibodies targeting neuron-specific antigens. The main IgG4 antibody-mediated neurological disorders (IgG4-ND) include MuSK myasthenia, autoimmune nodopathies with antibodies against nodal-paranodal cell-adhesion molecules (neurofascin-155 (NF155), contactin-associated protein 1 (Caspr1), and neurofascin isoforms (NF140/186), Morvan syndrome, or neuromyotonia, anti-LGI1- and CASPR2-associated limbic encephalitis, and several cases of the anti-IgLON5 and anti-DPPX-spectrum CNS diseases. However, because IgG4 antibody titers appear to be decreased in remission and increased in exacerbation, they may serve as potential biomarkers of treatment response, further supporting a pathogenic role for self-reacting B cells. Patients with autoimmune nodopathy usually have characteristic symptoms that emphasize the subacute onset of severe neuropathy, tremor, and sensory ataxia [69,70,71,72].

Most significantly, they respond poorly to IVIg and plasmapheresis, but excellently to rituximab, which induces long-term remissions. Although patients with anti-LGI-1 and CASPR2 antibodies are characterized by clinical heterogeneity, they also demonstrate considerable overlap in clinical symptomatology; anti-LGI1 antibodies are most commonly associated with epilepsy and limbic encephalitis, while anti-CASPR2 antibodies are associated with neuromyotonia, Morvan syndrome, and neuropathic pain. Anti-IgLON5 antibodies define a complex syndrome of chronic progressive brainstem symptomatology, gait instability, distinct non-rapid eye movement (REM) and REM parasomnias, sleep-disordered breathing, obstructive sleep apnea, cognitive decline, and movement disorders as recently identified, most commonly craniofacial dyskinesias, chorea, dystonia, and abnormal eye movements [3,71,72,73,74,75].

## 7. Treatment

The majority of patients with MuSK-MG have little or no therapeutic response to treatment with anticholinesterase drugs and experience an increase in cholinergic symptoms even at low doses [4]. Modoni et al. demonstrated that cholinergic hyperactivity to standard doses of acetylcholine esterase inhibitors (AchE-Is) is a relatively common symptom in patients with MuSK-MG, independent of AchE-I treatment, and may be an inherent feature of the disease [75]. In addition, the response to standard doses of pyridostigmine used in AchR-MG is ineffective and poorly tolerated due to its side effects. Among symptomatic medications for MuSK-MG, 3,4-diaminopyridine (3,4-DAP), ephedrine, and albuterol have recently been considered. The use of 3,4-DAP in patients with MuSK-MG has been described as moderately to mildly effective, with no notable side effects [5,76,77,78,79,80].

Conventional immunosuppressants are not commonly able to replace steroids in the maintenance of the satisfactory long-term control of symptoms. In MuSK-MG patients with exacerbated symptoms, high-dose prednisone, combined with plasma exchange, is recommended. Intravenous immunoglobulin should also be considered in these patients. In patients with contraindications to steroids, traditional immunosuppressive drugs (azathioprine, tacrolimus, mycophenolate, methotrexate, and cyclosporine) have been used, but achieving and ensuring long-term and complete symptom control is usually more difficult than in Ach-R MG patients. However, the majority of MuSK-MG patients are refractory to treatment. In these cases, the use of rituximab has shown promising results leading to sustained symptom control [80,81,82,83].

According to expert opinion, the treatment of MuSK-MG with rituximab (RTX), a monoclonal antibody directed against the CD20 receptor, is highly effective. RTX is more successful in MuSK-MG than in other MG subgroups and can be used for treatment at an earlier stage [84,85,86]. Rituximab is a chimeric murine/human monoclonal antibody produced by genetic engineering of Chinese hamster ovary tissue culture cells. It is a glycosylated immunoglobulin containing fixed sequences of human IgG1 and variable sequences of mouse light and heavy chains. It binds selectively to the transmembrane antigen CD20, which is found on the surface of B lymphocytes (circulating naïve and memory B cells) and is absent on other cells. RTX induces the death of cells containing the CD20 antigen by mechanisms dependent on both the complement system and those associated with antibody-dependent cellular cytotoxicity, as well as by apoptosis. B lymphocyte stem cells are devoid of the CD20 antigen, and the B lymphocyte population can be reconstituted after treatment with rituximab [85,86,87]. Marino et al. studied the long-term effects of RTX in nine treatment-resistant patients with MuSK-MG, with follow-up periods of 17 months to 13 years. Their data demonstrated that the therapeutic effects of RTX can continue for several years following treatment, suggesting that by depleting autoreactive B-cell clones, RTX can markedly disrupt the immunopathogenic circuitry responsible for maintaining the disease [83]. It is recognized that B-cell activity depends on T–B lymphocyte cross-talk and cooperation. Future studies are needed to investigate the effect of RTX on such interaction, particularly in relation to specific T- and B-cell repertoires.

Other antibodies targeting B lymphocytes (CD20, CD19) with potential relevance in myasthenia gravis, but without documented efficiency, include ocrelizumab, ofatumumab, obinutuzumab, ublituximab, and ibalizumab. In addition, the potential efficacy of the proteasome inhibitor was described in a report on the favorable impact of this drug in a patient with severe myasthenia gravis with anti-MuSK antibodies [88,89,90,91,92].

Promising therapeutic targets in patients with MuSK-MG are monoclonal antibodies against molecules involved in B-cell activation and against B cells at different stages of their maturation (e.g., plasmablasts). Precision medicine using chimeric autoantibodies against the T-cell receptor (CAAR-T) can also be effective. These are designed to target antigen-specific B cells in MuSK-MG. Other drugs are monoclonal antibodies against FcRn receptors: rozanolixizumab and efgartigimod. The principle of action of FcRn, a neonatal Fc receptor, is to bind to the Fc region and rescue IgG from acidic lysosomal degradation. This promotes recycling. The mechanism of action is very similar to that of IVIG. Efgartigimod is the first FcRn inhibitor to be approved for use in AChR-MG [93,94,95,96]. Clinical trials have shown that FcRn inhibitors are more effective in MuSK-MG patients than in AchR-MG patients. IgG4-dominant MuSK-MG is a weak complement-activating subclass of immunoglobulin, and complement inhibition is not effective [97,98,99]. Results from the REGAIN study showed that eculizumab was most effective in patients with anti-AChR antibodies [100,101]. Eculizumab is a humanized monoclonal antibody directed against the C5 component of the complement system. It inhibits the final step of complement activation and the formation of membrane attack complexes (MACs). It does this by blocking the conversion of C5a to C5b. With regard to the potential efficacy of the proteasome inhibitor bortezomib, at least one case of a beneficial effect of this drug has been described in a patient with severe myasthenia gravis with anti-MuSK antibodies [102,103].

In MuSK-MG, the thymus is usually atrophic. Thymic follicular hyperplasia (TFH) occurs in rare cases. They are characterized by a more severe course and less responsive to standard immunosuppression [104]. For patients with MuSK-MG, thymectomy is not currently recommended. However, anterior mediastinal imaging is required in all patients with an established diagnosis of MG, regardless of antibody type [105].

## 8. Conclusions

MuSK myasthenia gravis is a more aggressive and difficult-to-treat form of neuromuscular junction disease. In recent years, new information has become available on the potential pathomechanisms of this form of MG. Advances in research into immunopathogenesis will contribute to the correct diagnosis of this autoimmune disease and the application of effective treatment. Further research is needed on the role of thymus in MuSK MG pathogenesis and its role as a potential therapeutic target too.

## Figures and Tables

**Figure 1 cells-13-00556-f001:**
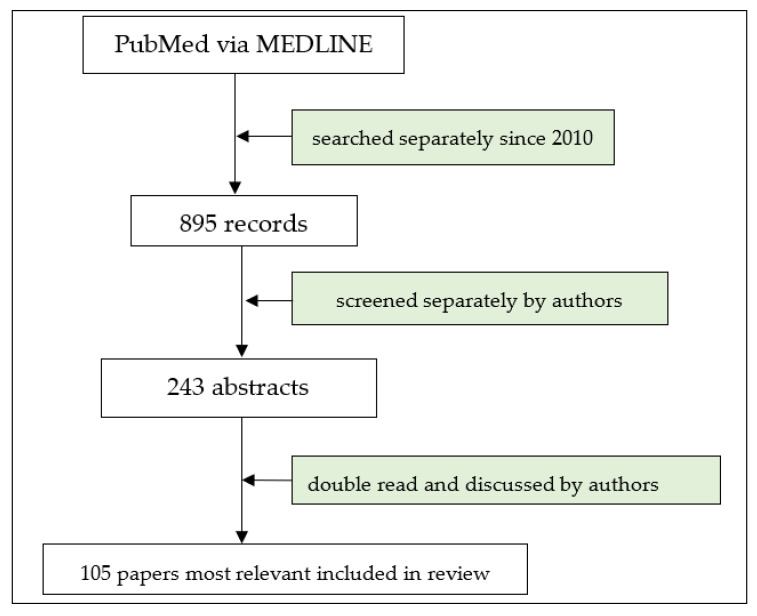
Flow chart of study selection.

**Figure 2 cells-13-00556-f002:**
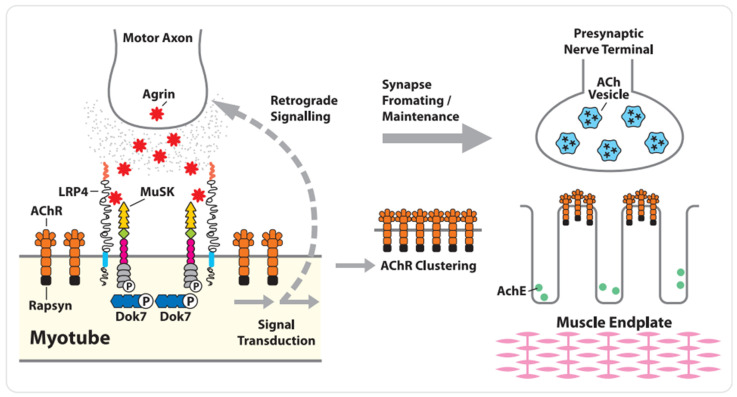
Developing NMJ: The key proteins in NMJ formation include a neuronally derived heparan-sulfate proteo-glycan, agrin, and three muscle proteins: downstream of kinase-7 (Dok7), low-density lipoprotein receptor-related protein-4 (LRP4), and rapsyn [13,14]. The low-density lipoprotein receptor-related protein-4 (LRP4) serves as a cis-acting (in muscle) transmembrane ligand for MuSK; agrin acts as an allosteric regulator of LRP4 interaction with MuSK; downstream of kinase-7 (Dok7) functions as a cytoplasmic activator of MuSK, whereas rapsyn binds directly to AChR to facilitate its clustering (based on [9], own modification).

**Figure 3 cells-13-00556-f003:**
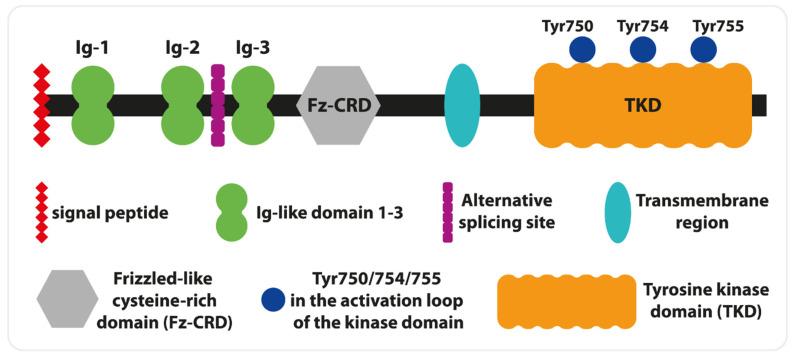
MuSK Structure (modified from [16]).

## Data Availability

Not applicable.

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
