# Peer review of "MuSK Myasthenia Gravis—Potential Pathomechanisms and Treatment Directed against Specific Targets"

_cells, 2024, doi:10.3390/cells13060556_

Round 1
Reviewer 1 Report
Comments and Suggestions for Authors
This is a nice review of the current literature of MuSK MG.
There are some minor issues that can be improved:
- It is unclear why there is are different writings of Myasthenia gravis (myasthenia gravis vs MG).
- The literature search is a clear advantage of the paper. Please also add a prism chart for a better overview of the process.
- Please expand on the paragrah on FcRn inhibition. As far as I am concerned, this strategy could be valuable for MuSK patients. Please explain why FcRn inhibtion, as opposed to complement inhibiton, might work for MuSK MG and the current studies of FcRn inhibition that may also include MuSK patients.
Author Response
Dear Editor,
First of all we would like to thank the Editors and Reviewers for the time taken in reviewing our study. We highly value all the comments and truly believe that implementing these changes into our paper will improve the final manuscript. Below you will find a point-by-point response: we copied the sections that were modified in the main text between squared brackets, and noted pages where changes were made. Changes to the main document are highlighted in yellow.
On behalf of the authors,
Marta Waliszewska-ProsóÅ‚
Reviewer 1
- It is unclear why there is are different writings of Myasthenia gravis (myasthenia gravisvs MG).
Thank you for this comment. It was corrected
- The literature search is a clear advantage of the paper. Please also add a prism chart for a better overview of the process.
Thank you for this comment. A prism chart has been added.
- Please expand on the paragraph on FcRn inhibition. As far as I am concerned, this strategy could be valuable for MuSK patients. Please explain why FcRn inhibition, as opposed to complement inhibition, might work for MuSK MG and the current studies of FcRn inhibition that may also include MuSK patients.
Thank you for this comment. We have tried to expand the section on drugs that work by this mechanism.
Reviewer 2 Report
Comments and Suggestions for Authors
There is too much detail about the diagnostic tests, especially SFEMG - please reduce and cite relevant sources.
A propos CNS manifestations of MuSK MG, where is Musk found in the brain, and do these manifestations improve when the MG improves?
Rituximab is recommended for MuSK MG in the International Consensus Guidance statements - this should be mentioned and cited.
Author Response
Dear Editor,
First of all we would like to thank the Editors and Reviewers for the time taken in reviewing our study. We highly value all the comments and truly believe that implementing these changes into our paper will improve the final manuscript. Below you will find a point-by-point response: we copied the sections that were modified in the main text between squared brackets, and noted pages where changes were made. Changes to the main document are highlighted in yellow.
On behalf of the authors,
Marta Waliszewska-ProsóÅ‚
Reviewer 2
- There is too much detail about the diagnostic tests, especially SFEMG - please reduce and cite relevant sources. A propos CNS manifestations of MuSK MG, where is Musk found in the brain, and do these manifestations improve when the MG improves?
Thank you for this comment. Interestingly, two MuSK isoforms (one identical to the skeletal muscle variant) are expressed in the brain. It was shown that hippocampal MuSK isoforms play a crucial role in cholinergic response and help memory formation [https://doi.org/10.1111/ejn.15382]. These data obtained on animal models confirm observations of MG patients with memory deficits [DOI: 10.1007/s004150050139].
- Rituximab is recommended for MuSK MG in the International Consensus Guidance statements - this should be mentioned and cited.
Thank you for this suggestion. Required information has been added.
Reviewer 3 Report
Comments and Suggestions for Authors
I read with interest the review of Dziadkowiak and collegues. The work has a defined structure that guide the authors throug the text, even if sometimes it is not completely clear what is the messagge that the authors wants to convey.
I have some observation that could improve the manuscript:
at the line 93 it is reported that MuSK has different isoforms: it would be helpful to state if these isoforms are expressed in different tissues.
at the line 94 it is written "MUSK" insetad of "MuSK".
line 108-109 the sentence is not clear: please reconsider it.
line 156 the whole paragraph is not clear. In particular "NCBI data indicate..." what do the author mean? Clarify which database is used. Please revise the paragraph.
line 206 to the end of the paragraph: it is not completely clear what is the messagge. Can the authors somehow summarize the conflicting observation?
I would also reccomend to rewrite the conclusion section: it seems to short and it should give to the reader a final messagge to take home.
More general, the role of tymus is partially mentioned. Do the authors consider that thymus could be further investigated in MuSK MG pathogenesis and can it has a role as potential therapeutic target?
Author Response
Dear Editor,
First of all we would like to thank the Editors and Reviewers for the time taken in reviewing our study. We highly value all the comments and truly believe that implementing these changes into our paper will improve the final manuscript. Below you will find a point-by-point response: we copied the sections that were modified in the main text between squared brackets, and noted pages where changes were made. Changes to the main document are highlighted in yellow.
On behalf of the authors,
Marta Waliszewska-ProsóÅ‚
Reviewer 3
- At the line 93 it is reported that MuSK has different isoforms: it would be helpful to state if these isoforms are expressed in different tissues.
Thank you for this comment. We were trying to rebuilt this part according to available literature.
- At the line 94 it is written "MUSK" instead of "MuSK".
Thank you for this comment. It has been corrected.
- Line 108-109 the sentence is not clear: please reconsider it.
Thank you for this comment. We have rewritten this section.
- Line 156 the whole paragraph is not clear. In particular "NCBI data indicate..." what do the author mean? Clarify which database is used. Please revise the paragraph.
Thank you for this comment. It was changed for: According to the National Centre for Biotechnology Information data (NCBI, www.ncbi.nlm.nih.gov), 52,922 different mutant variants of the MUSK gene were detected.
- Line 206 to the end of the paragraph: it is not completely clear what is the message. Can the authors somehow summarize the conflicting observation? I would also recommend to rewrite the conclusion section: it seems to short and it should give to the reader a final message to take home. More general, the role of thymus is partially mentioned. Do the authors consider that thymus could be further investigated in MuSK MG pathogenesis and can it has a role as potential therapeutic target?
Thank you for this comment. We have rewritten this section.
Round 2
Reviewer 3 Report
Comments and Suggestions for Authors
The authors revised the manuscript as suggested. I belive that it is now in a form suitable for publication.